# Exploring the Link Between Infections and Primary Osteoarthritis: A Next-Generation Metagenomic Sequencing Approach

**DOI:** 10.3390/ijms26010020

**Published:** 2024-12-24

**Authors:** Irina Niecwietajewa, Jakub Banasiewicz, Gabriel Zaremba-Wróblewski, Anna Majewska

**Affiliations:** 1Department of Medical Microbiology, Medical University of Warsaw, Chalubinski 5 Str., 02-004 Warsaw, Poland; 2Department of Trauma and Orthopedic Surgery, Czerniakowski Hospital, 19/25 Stępińska St., 00-739 Warsaw, Poland; 3Department of General, Vascular and Oncological Surgery, Medical University of Warsaw, 02-091 Warsaw, Poland

**Keywords:** *Yersinia enterocolitica*, *Escherichia coli*, *Synechococcus*, primary knee osteoarthritis

## Abstract

This prospective pilot study examined the association between microorganisms and knee osteoarthritis by identifying pathogens in the synovial membrane, synovial fluid, and blood samples from two patients with primary bilateral knee osteoarthritis, using metagenomic next-generation sequencing (mNGS). Intraoperatively, during routine knee arthroplasty procedures, we collected the following 12 samples from each patient: two synovial membrane samples, two synovial fluid samples, and two venous blood samples. After DNA isolation and library construction, each sample was subjected to deep whole-genome sequencing using the DNBSEQT17 platform with the read length PE150 as the default. Metagenomic sequencing data were mapped to the NCBI NT database to determine species abundance. The predominant species in all samples tested were classified under the Enterobacterales order, the most abundant being *Yersinia enterocolitica.* The second and third most common microorganisms detected were *Escherichia coli* and autotrophic, Gram-negative bacteria *Synechococcus* sp., which is a bioaerosol component, indicating a risk of inhalation of the toxic metabolites of this latter microorganism. This article provides an initial exploration of mNGS use to study the etiopathogenetic mechanisms of knee osteoarthritis (OA). While our analysis identified bacterial DNA, particularly from *Yersinia*, further cross-sectional studies in larger populations with and without OA are needed to determine the role of these agents in OA pathogenesis.

## 1. Introduction

Osteoarthritis is a leading cause of disability worldwide, especially as populations age. This disorder is currently a leading cause of disability in older people, and it is predicted to be the most common cause of disability in the general population by 2030. If the current trend continues, it is estimated that by the year 2050, almost 1 billion people will have some form of osteoarthritis. As the world’s population ages, the health and economic burden associated with osteoarthritis will continue to increase. The World Health Organization has designated 2021–2030 as the Decade of Healthy Aging, emphasizing not only life expectancy but also quality of life. This initiative provides a valuable opportunity to address the burden of OA in the broader context of adult health, particularly given the chronic nature of the condition and its impact on mobility and daily activities [1].

OA is characterized by the localized destruction of articular cartilage and full-thickness cartilage loss. Originally thought to be a disease of the articular cartilage alone, it is now known that other tissues, such as the synovial membrane and subchondral bone, are also involved in the development of OA, which can produce inflammatory mediators and induce cartilage degradation [2].

Clinical symptoms of OA include pain, tenderness, limited range of motion in the affected joint, crepitus in the affected joint, stiffness after activity, and joint swelling. On radiological images, degenerative changes can be seen in the form of joint space narrowing, subchondral bone sclerotization, cysts, and osteophytes in the region of the affected joint [3,4].

OA occurs in both primary and secondary forms. The pathogenesis of secondary OA is easier to define—it results from various joint-damaging factors in patients, including injuries to the cartilage or bone, dysplasia, infections, or autoimmune diseases. The causes of primary OA, which develops in previously undamaged joints, remain unclear. Many investigators believe that the daily stresses on the joints play an essential role in the development of OA, which is a so-called “disease of use”. External forces acting on the tissues accelerate the catabolism of chondrocytes and destroy the matrix, leading to cartilage loss [4,5]. In addition to overuse during sports and everyday life, factors contributing to the progression of OA include older age, being overweight or obese, female sex, genetic factors, joint injury, work/environmental hazards, abnormal joint morphology, and malalignment [3,4].

Advances in molecular biology have changed our understanding of the origin of many diseases. The discovery of the involvement of inflammatory mediators and inflammation-related genes in the pathomechanism of OA has led to the emergence and strengthening of the concept of inflammation as a core disease process [6]. This mechanism is activated by the recognition of pathogen-associated molecular patterns (PAMPs), which include bacterial and viral ligands, and their binding to pattern recognition receptors (PRRs) [7]. PRRs include Toll-like receptors (TLRs), which are the primary signaling receptors of the innate immune system. Kim et al. showed that the expression of TLRs is increased in articular cartilage during OA [8]. TLR-2 and TLR-4 ligands, such as low molecular-weight hyaluronic acid and fibronectin, have been detected in the synovial fluid of joints with OA [9,10]. Each type of TLR binds the antigens of specific pathogens: TLR2 binds Gram-positive and Gram-negative bacteria, while TLR 4 binds Gram-negative bacteria and *Candida albicans* [11]. Those phenomena occur early in the disease, and innate immunity initiates and causes damage to joint structures through low-grade inflammation. Nair et al. showed that analysis of synovial fluid from patients with early cartilage damage in OA indicated an increased synoviocyte response to TLR-2 and TLR-4 ligands [10]. In addition, the increase in IL-15, a cytokine released in response to TLR-4 receptor activation, occurs early in OA, suggesting a potential role for the innate immune system in the pathogenesis of OA [12].

The introduction of the term reactive arthritis (ReA) for arthritis observed after a bacterial infection in which “no microorganisms are isolated from the synovial fluid” was a historical milestone in the classification and inclusion of ReA in the group of rheumatic diseases. ReA is an inflammatory form of joint injury classified as a seronegative spondyloarthropathy with frequent HLA-B27 genetic predisposition. ReA is a condition of aseptic synovitis that often develops 2–4 weeks after distant (extra-articular) infection with bacteria such as *Chlamydia*, *Salmonella*, *Shigella*, *Escherichia*, *Campylobacter*, and *Yersinia* [13]. ReA is also associated with urogenital and other infections [14]. Furthermore, most microorganisms can cause asymptomatic or chronic infections that are not always recognized. To date, no link has been established between primary OA and bacterial infections; however, correlations have been found between OA and gut microbiota [15]. Next-generation metagenomic sequencing (mNGS) appears to be the most suitable tool to determine a pathogen’s role in the etiology of a disease. mNGS can detect bacterial DNA that comprises less than 1% of the total DNA in a sample and represents a comprehensive approach to identifying numerous pathogens, including viruses, bacteria, fungi, and parasites, in a single test. This approach has been successfully validated for the diagnosis of several infectious diseases [16,17].

### Objective

This study aims to determine the association between microorganisms and knee OA by identifying pathogens in the synovial membrane, synovial fluid, and blood samples from two patients with bilateral primary knee osteoarthritis (PKOA), using next-generation metagenomic sequencing. Due to the limited number of patients and biological samples included, our aim is not to prove a cause-and-effect relationship but to reignite interest in the infectious and inflammation background of OA.

## 2. Results

### 2.1. Taxonomic Distribution

In the samples collected from patients, microorganisms classified into the following phyla were predominant (based on the detectable reads): Proteobacteria (75.1% and 73.1% in patients A and B, respectively); Cyanobacteria (16.1% and 13.9% in patients A and B, respectively); Ascomycota (6.2% and 7.9% in patients A and B, respectively); and Firmicutes (1.9% and 4% in patients A and B, respectively). The predominant species in all samples tested was *Y. enterocolitica* (family *Yersiniaceae*, order Enterobacterales, class Gammaproteobacteria) with the following mean relative abundance in the samples: patient A—blood samples 47.158, synovial fluid 43.641, synovial membrane 46.489; patient B—blood samples 44.440, synovial fluid 39.607, synovial membrane 43.776.

The second most abundant species were *E. coli* (family *Enterobacteriacae*, order Enterobacterales, class Gammaproteobacteria) and *Synechococcus* sp. (family *Synechococcaceae*, order Synechococcales, class Cyanophyceae). The top 30 species, according to the relative abundance of microorganisms in blood samples, synovial fluid samples, and synovial membrane samples, are shown in Figure 1.

Multi-level data for the microorganisms detected in the patient samples were visualized on Krona diagrams provided in the Appendix A.

### 2.2. Microbial Communities

The study also focused on the variation in the abundance of microbial communities. A heatmap (Figure 2) was used to show the distribution of species in individual patients and the similarity between patients and species through clustering. The heatmap displays the top 30 most highly abundant microorganisms (in descending order). In both patients, *Y. enterocolitica*, *E. coli*, and *Synechococcus*_sp_PROS-9-1 predominated in the clustered samples.

In samples from patient A, the DNA of 95 species of microorganisms was identified. In samples from patient B, the DNA of 128 species of microorganisms was identified. In both patients (A, B), 83 of the same microbial species were identified. In patient A, 12 additional species were identified with the highest relative abundance for the species *Curtobacterium flaccumfaciens* (in tissues) and *Fusarium pseudograminearum* (in synovial fluid). In patient B, 45 species were identified that were not found in samples from patient A. The species with the highest relative abundance, in order, were as follows: *Campylobacter ureolyticus* (blood and fluid) and anaerobic bacteria (*Clostridium perfringens* and *Clostridium bornimense*, *Bacteroides fragilis, Prevotella melaninogenica*). The similarities and differences in species composition are shown in Figure 3.

Figure 4 also shows an analysis of differences in species abundance between the patients. Overall, 30 species with the highest relative abundance were included in the analysis. The following organisms exhibited differences in abundance between the two patients: *Y. enterocolitica*, *Synechococcus* spp., and *Burkholderia pseudomallei*. 

### 2.3. Taxonomic Diversity

Figure 5a shows the alpha diversity within individual samples (blood, synovial fluid, and synovial tissue) from both patients; Figure 5b shows the alpha diversity within samples collected from individual patients; Figure 5c illustrates the beta diversity of homogeneous samples obtained from both patients; and Figure 5d shows the differences in species diversity between patients.

The similarity index and distance between samples are shown in Figure 6.

The gene diversity analysis is shown in the Appendix A shows gene alpha diversity analysis. Appendix A present gene beta diversity. 

## 3. Discussion

Participants in the study group were selected by carefully screening all patients scheduled for a total knee alloplasty. Two eligible individuals met the following criteria for ‘true primary’ knee OA: no significant joint damage according to both their medical history and physical examination.

The microbiome in the collected tissues was evaluated using mNGS, which detects tiny amounts of DNA in samples and enables the identification of most known microorganisms, including those that cannot be cultured via the classical method.

Sequencing and bioinformatic analysis identified DNA from a wide range of microorganisms in the collected samples. The selection of the appropriate technology from the variety of sequencing systems available depends on the goal of the project [18]. We used the DNBSEQ platform to achieve our goal. DNBSEQ technology sets a new benchmark for NGS accuracy through the innovative application of DNA nanoballs (DNBs) and rolling circle replication (RCR). This method significantly reduces errors during DNA amplification, ensuring reliable detection of genetic variants even at low frequencies. DNBSEQ-T7 is suitable for WGS metagenomics, generating up to 6Tb of high-quality data with read lengths of 150 bp. The frequency of index misassignment of the DNBSEQ platform, which uses a combinatorial probe-anchor synthesis method and DNA nanoball sequencing technology developed by MGI, was shown to be as low as 0.0001–0.0004% [18]. Therefore, in our analysis, we considered microorganisms to be present with more than 10% of non-human DNA reads in the sample.

Both patients had positive mNGS results. The main pathogenic microorganisms identified were Gram-negative bacilli, namely, *Y. enterocolitica*, followed by *E. coli* and *Synechococcus* spp. Among the species with a relative abundance of >10% in the collected samples were *Candidatus* Portiera aleyrodidarum, *Fusarium oxysporum*, *Burkholderia pseudomallei*, *Klebsiella pneumoniae*, and *Staphylococcus aureus*. 

The most abundant species in all samples from both patients were Gram-negative rods belonging to the order Enterobacterales, specifically *Y. enterocolitica*, *E. coli*, and *K. pneumoniae*, accounting for 66% of the reads in patient A and 64% in patient B. 

As recent reports have shown, TLR CD14-lipopolysaccharide-binding protein (LBP) is central to the pathogenesis of low-grade inflammation relevant to the development of OA [19].

Lipopolysaccharide (LPS) is a component of the Gram-negative bacterial cell wall and a bacterial endotoxin. LPS is a potent activator of the inflammatory response and is a crucial PAMP in Gram-negative bacteria. Even a small amount of LPS present in the blood during an infection is sufficient to trigger an inflammatory response through interactions with TLRs [20]. LPS activates TLR4, leading to the production of numerous pro-inflammatory cytokines and, thus, systemic low-grade inflammation [20,21].

The involvement of LPS in the pathomechanism of cartilage destruction by low-grade inflammation has been demonstrated in animal models of trauma-related injured and uninjured joints [22,23].

*Y. enterocolitica* was the predominant species in all samples from both patients with advanced PKOA. Yersiniosis is most commonly reported food-borne zoonosis in Europe, and it is usually associated with conditions such as mesenteric lymphadenitis, pseudomembranous appendicitis, and even inflammatory bowel disease. Infections can range from asymptomatic to life-threatening sepsis. Its transmission to humans usually results from consuming contaminated milk, water, pork, or tofu. Once it has entered the gastrointestinal tract, *Y. enterocolitica* can invade epithelial cells and penetrate through mucous membranes within the ileum, which is followed by replication within Peyer’s patches. Outside the GI tract, it is also capable of causing ReA, especially in individuals with the human leukocyte antigen (HLA)-B27. While its epidemiology shows that *Y. enterocolitica* has a global distribution, it is often found in colder climates, including those of northern Europe and Japan [14,24].

Since the 1980s, reports have been published on extraintestinal disorders caused by *Y. enterocolitica*, primarily affecting joints and other organs. Furthermore, a potential link between inflammatory and osteoarthritic diseases and previous infection with *Y. enterocolitica* and possibly other Gram-negative bacteria was postulated [25].

As early as 1989, it was shown that patients with ReA after *Yersinia* infections were positive for *Yersinia* antigens in the synovial fluid of their affected joints [26]. Furthermore, *Yersinia* LPS has been detected in the leukocytes of patients with ReA caused by *Y. enterocolitica* O:3 even four years after infection [27].

Although ReA is considered a sterile disease with no microorganisms found in the joints, immune complexes with *Yersinia* antigens, *Yersinia*-specific antibodies of the IgM, IgG, and IgA classes, and LPS have been found in serum and synovial fluid from patients diagnosed with *Yersinia*-induced ReA, even several years after infection [28]. The strong arthritogenic potential of *Y. enterocolitica* has also been demonstrated in studies on animal models [29,30,31]. Ultimately, *Y. enterocolitica* serotype O:3 was identified as the leading cause of arthritis. Bacteria in the blood enter the joint tissues via plasma or lymphoid cells. After eradication of the infection, cellular components of *Y. enterocolitica* may persist in the affected joint. Kasperkiewicz et al. demonstrated the presence of antibodies recognizing *Yersinia* LPS in synovial fluid from patients with juvenile idiopathic arthritis (JIA) [21]. Results from Norway show an association between yersiniosis and diseases caused by chronic inflammation such as inflammatory joint disease, ankylosing spondylitis, rheumatoid arthritis, and iridocyclitis [32]. A publication from China describes a case of spontaneous bloodstream infection with an etiology of *Y. enterocolitica* in a 56-year-old patient with ankylosing spondylitis with no history of gastrointestinal infection [33].

Notably, the patients in our study group also reported no history of acute gastrointestinal infections. In addition, patient A had a history of multi-level spondyloarthritis and had undergone surgery for lumbar spinal stenosis years ago.

Despite the fact that previous studies have shown the presence of cellular elements of *Y. enterocolitica* in joint tissues, and that DNA of *Y. enterocolitica* dominated in samples from both patients with advanced PKOA in our study, it does not indicate a direct association of this species with damage to joint structures, but *Y. enterocolitica* should be afforded special attention.

*E. coli* was the second most common microorganism identified in the samples from the participants in our study. Findings showing an association between inflammatory joint diseases and *E. coli* infection are much less numerous than for *Y. enterocolitica*, and they mainly relate to ReA [34], rheumatoid arthritis [35], ReA [36] and experimental arthritis in response to the immunization of rats with *E. coli* LPS [37].

The third most common microorganism found in the samples was *Synechococcus* spp. These autotrophic cyanobacteria are coccoid or rod-shaped microbes with a diameter of <2 μm. These bacteria are found in aquatic environments worldwide and are one of the most common microbes [38]. They are also bioaerosol components, which gives rise to risk of inhalation of the toxic metabolic products of *Synechococcus* spp. (e.g., in the Baltic Sea region) [39,40].

*Candidatus* Portiera aleyrodidarum, the fourth most common species found in the samples, is an obligate endosymbiotic bacterium that resides in the digestive tract of whiteflies, including *Bemisia tabaci*, which is one of the most damaging pests of crops and one of the 100 most invasive species in the world [41].

The fifth most frequently detected microorganism, *Fusarium oxysporum*, are ubiquitous molds that cause food spoilage, primarily of bananas [42]. They are also responsible for a broad spectrum of human infections, including superficial, invasive, and disseminated infections. The clinical picture depends on the host’s immune status and the fungal invasion site [43,44].

*Burkholderia pseudomallei*, the sixth most common microorganism identified, is a Gram-negative, intracellular bacterium living in soil in tropical zones. *B. pseudomallei* causes melioidosis, which is transmitted through the skin or ingestion or inhalation of contaminated soil or water. It can cause a broad spectrum of conditions. Although *B. pseudomallei* is commonly considered an endemic microorganism of southeast Asia and northern Australia, there is increasing evidence that it is likely to be endemic in many tropical areas and that soil and climatic conditions favorable for *B. pseudomallei* also occur in other regions around the world [45].

Interpreting mNGS results, especially detecting DNA of low biomass, presents a significant challenge. The method’s very high sensitivity makes it prone to detecting contaminants and DNA from microorganisms that do not originate from the sample, the source of which may be the environment, personnel, instruments, and laboratory equipment. Sample contamination can occur at any stage of the testing process such as sample collection and processing or during the sequencing process [46]. In one conducted study, sample collection and processing were carried out under controlled conditions. Samples were collected in an operating theater setting with all procedures in place to exclude contamination. Airborne microorganisms present during the surgical procedure were not identified in the sequencing. Each step of the study was carried out in dedicated rooms, under controlled environmental conditions, and by personnel protected with protective measures to prevent contamination inside the laboratory. The procedure for each sequencing step included rigorous quality controls according to laboratory standards.

At the library construction stage, each sequence derived from the samples was uniquely barcoded to exclude random non-derived sequences, which may represent contaminants, from further analysis [47]. Moreover, at the filtering stage, the raw sequencing data often contain a certain proportion of uncertain bases, low-quality sequences and adapter sequences. To ensure the reliability of subsequent analysis results, the data were filtered using SOAPnuke software to obtain high-quality clean data.

Chrisman et al. also highlight the possibility of computational errors during data analysis. There is a risk of misalignment of uncertain (poorly matched) and unmapped reads to the taxonomic units of different microorganisms, including viruses, bacteria, and archaea, by Kraken 2, which is commonly used in bioinformatic analysis. The errors analyzed by the authors are due to, among other things, the commonly used human reference genome GRCh38, which is incomplete: it lacks large heterochromatic sequences and, as a single, linear reference genome, does not represent the full spectrum of human genetic diversity. The deficiencies present in GRCh38 were filled in the newly developed reference genome T2T-CHM13, which was used in our study [48,49].

Nevertheless, almost all of the microorganisms identified in our study in the joint tissues and blood of both patients with PKOA, according to Chrisman et al., belong to the top 100 most abundant microorganisms that were most common in the samples analyzed for poorly matched reads, which were classified by Kraken 2 into microbial species. However, the authors of the aforementioned study allow for the possibility that these bacteria are part of the natural blood microbiome [48].

### The Limitations of the Study

First, given this is a pilot study designed to guide further research, the limited number of carefully selected patients included represents a limitation of the study. In addition, it is recognized that there is a need to include samples from healthy individuals (with no clinical or radiological signs of OA) and to take a thorough history of the likelihood of yersiniosis as well as to expand the group of patients with OA to at least 50 individuals.

Second, the results were not verified by the direct confirmation of bacteria in clinical samples. The presence of bacterial DNA in joint tissues in individuals with osteoarthritis (OA) does not unequivocally confirm a role for bacteria in the initiation and progression of lesions.

Finally, mNGS is an advanced technique that accurately identifies the DNA of etiological agents responsible for the disease. However, owing to its high sensitivity, the results may be inconclusive and difficult to interpret at this level of knowledge. Future studies would benefit from the inclusion of real-time PCR technique for the detection of specific microbial sequences in clinical samples with determination of threshold cycle, in parallel with mNGS analysis, including the functional analysis. Another tool for understanding the pathogenesis of OA may be the use of animal models. The induction of infection and inflammation in animals will not only help to understand intracellular changes and processes but will also help to establish the next milestones in the understanding of this complex pathology.

## 4. Materials and Methods

A prospective pilot study was conducted at the Department of Trauma and Orthopedic Surgery, Warsaw General City Hospital, from 1 September 2023 to 31 January 2024. The department performs more than 650 total joint arthroplasties annually, including approximately 300 knee arthroplasties.

In accordance with its definition, a pilot study is the first step of research protocol. It is a small-sized study which assists in the planning and modification of the main study.

### 4.1. The Patients

The study included carefully selected patients from the hospital’s orthopedic clinic diagnosed with PKOA and eligible for total knee arthroplasty. 

The study group included 2 participants (designated A and B) diagnosed with bilateral PKOA who met the following inclusion criteria:-No medical history of symptoms suggestive of acute arthritis, including reactive post-infectious arthritis;-No medical history of musculoskeletal trauma;-No autoimmune or metabolic inflammatory joint disease: rheumatoid arthritis, morbid obesity, gout, diabetes, or psoriasis;-No active infection elsewhere in the body;-No active participation in sports that chronically stress the knee joint;-No neurodegenerative diseases;-Below the age of 70 years.

Table 1 provides detailed information about the study participants, including their demographic features and medical history. Preoperative laboratory test results and course of treatment are shown in the Appendix A.

Figure 7 shows radiographs of both study participants taken before knee allograft surgery.

### 4.2. Samples

To identify microbial DNA in fluids and tissues taken from the patients with PKOA, samples were collected from the peripheral venous blood (b), synovial fluid (f), and synovial membrane (t) of the knee joint prior to endoprosthesis implantation. All samples were collected intraoperatively during the routine steps of knee allograft surgery in a manner that did not affect the duration of the procedure or cause additional tissue damage. The samples were collected from patient A on 25 October 2023 and from patient B on 10 November 2023.

In detail, 6 mL of peripheral venous blood was collected into an EDTA vacuum tube (EDTA represents anticoagulant ethylenediaminetetraacetic acid) by the anesthesiologist through a short peripheral intravenous cannula that is routinely inserted prior to each surgical procedure before the patient is anesthetized. The operator collected 5 mL of joint fluid after incision and dissection of the skin and subcutaneous tissues superficial to the knee joint. The joint fluid was aspirated prior to making the incision in the joint capsule. Fragments of the synovial membrane, approximately 10 × 10 mm in size, were collected from the suprapatellar recess after dissection of the Hoffa’s body and dislocation of the patella with flexion of the knee joint.

To increase the reliability of metagenomic sequencing, each sample was divided into two separate samples immediately after being collected in the operating room. 

After aspiration, the synovial fluid was transferred to two 2 mL DNase- and RNase-free sterile centrifuge tubes (Axygen, Middlesex, UK). After collection, the synovial membrane fragments were rinsed with sterile saline, dried, and then debrided of any remaining pieces of other tissues (fat and connective tissue). The cleaned synovial membrane was then cut into fragments of approximately 50 mg and divided equally into two 2 mL DNase- and RNase-free sterile centrifuge tubes (Axygen, UK). All samples were immediately placed on dry ice and transferred to the BGI TECH SOLUTIONS laboratory (Warsaw, Poland) for DNA isolation.

To prevent the contamination of samples, all stages of the study were carried out in dedicated rooms in a controlled environment using appropriate decontamination procedures and protective clothing (mask, clean suit, and gloves). Before starting the knee arthroplasty procedures in both patients, an air sample of 1 m^3^ was taken, using the AIR IDEAL^®^ 3P^®^ (bioMérieux, Lyon, France), for microbiological examination. After incubation according to standard procedures, the following results were obtained: patient A: total microbial count, 3 CFU (colony-forming units) per cubic meter (CFU/m^3^) of air, including *Staphylococcus epidermidis* (1 CFU/m^3^) and *Micrococcus* spp. (2 CFU/m^3^); patient B: total microbial count, 2 CFU/m^3^ of *S. epidermidis*.

### 4.3. DNA Isolation and Library Construction

DNA was extracted using the DNeasy Blood & Tissue Kit (Qiagen, 69504, Hilden, Germany). Before constructing the DNA library, the concentration, integrity, and purity (absence of protein, RNA, and other contamination) were evaluated. The DNA concentration was quantified using the Qubit dsDNA HS Assay Kit with a Qubit 4 Fluorometer (Invitrogen, Singapore). The concentration, mass, and DNA quality obtained during sample isolation are shown in the Appendix A.

DNA libraries were constructed and sequenced in accordance with the standard protocol established by the Beijing Genomics Institute (BGI Genomics, Shenzhen, China) for the DNBSEQ sequencing platform. Library construction and library quality control are described in the Appendix A.

### 4.4. Sequencing

The prepared library has been utilized for sequencing on the DNBSEQ-T7 platform with a default read length of PE150. The insert fragment size was observed to be within the range of 300 to 400 base pairs. The Phred + 33 Fastq Quality score was employed for the assessment and control of the sequence data. The DNBSEQ sequencers use rolling circle amplification (RCA) technology. Each amplification utilizes the original circular DNA template to circumvent amplification errors at the same position, thereby reducing the amplification error and enhancing accuracy.

### 4.5. Quality Control of Sequencing Reads and Removal of Host Reads

The initial raw data set was filtrated using the SOAPnuke software (version 2.2.1) to obtain a high-quality, clean data set. Reads containing ≥0.1% uncertain bases (N bases), reads containing adapter sequences, reads containing barcode sequences, and reads that contained more than 50% of low-quality bases (bases with Q < 20) were excluded from further analysis. The human host sequences were removed by aligning them to the human reference genome T2T-CHM13 v2.0 using Bowtie2 (version 2.4.4). Data filtering raw data and quality control results are presented in the Appendix A.

### 4.6. Assembly and Gene Prediction

The MEGAHIT software (version 1.2.9) was used to assemble the clean data. The de novo prediction of metagenomic genes was conducted using MetaGeneMark (version 3.38). The gene prediction results for each sample were subjected to redundant processing by CD-HIT (version 4.8.1). Once a non-redundant gene set had been constructed, TPM (Transcripts Per Million) was employed to quantify the relative abundance of different genes. Gene abundance was quantified using Salmon (version 1.6.0). The assembly results and gene prediction results are presented in the Appendix A.

### 4.7. Taxonomic Annotation

This study employed Kraken2 software and an NCBI NT database for species annotation. The sample sequences were aligned using the retrieved data from 6 May 2022 to calculate the number of sequences belonging to different species. The Bracken2 (version 2.6.1) program was employed to ascertain the actual abundance of species in the sample, thereby facilitating completion of the taxonomic annotations.

### 4.8. Diversities

The diversity and similarity of specimens were determined according to the following comparison plans. Plan 1. Differences between different specimen types without consideration of interpatient variability. Plan 2. Differences between patient A and patient B without considering the variability between sample types.

#### 4.8.1. Alpha Diversity

Alpha diversity analysis of gene and species abundance in samples and/or groups was quantified using the following indices: The Chao1 index quantifies the richness of genes and species. Pielou evenness, also known as Shannon’s index, measures the combined richness and evenness of genes and species. Furthermore, Simpson’s evenness index can be used for measuring the richness and diversity of gene and species occurrence.

#### 4.8.2. Beta Diversity

Beta diversity, defined as the distance and similarity between samples, was quantified using the Bray–Curtis distance.

### 4.9. Differential Species Analysis

The Wilcoxon signed rank test and Kruskal–Wallis test were employed to identify statistically significant differences between individual samples, sample types, and patients. A statistically significant relationship between variables was defined as a *p*-value of <0.05.

## 5. Conclusions

In the present study, the detection of *Y. enterocolitica* in almost half of the reads in all synovial membranes, synovial fluid, and blood samples may suggest a potential association between infection with this bacterium and PKOA. This article provides an initial exploration of mNGS use to study the etiopathogenetic mechanisms of knee OA. While our analysis identified bacterial DNA, particularly from Yersinia, further cross-sectional studies in larger populations with and without OA are needed to determine the role of these agents in OA pathogenesis.

## Figures and Tables

**Figure 1 ijms-26-00020-f001:**
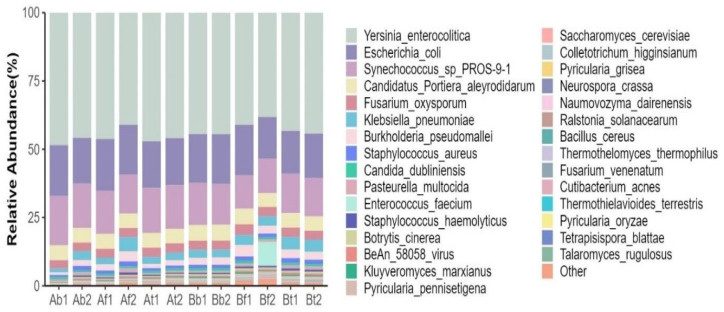
The species abundance columnar stack graph. The ordinate represents the relative abundance of the species; the abscissa represents the samples taken from 2 patients (patient A and patient B; b represents the blood sample, f—synovial fluid, t—tissue, synovial membrane). Two samples (1 and 2) of the same biological material were taken from each patient. The column color represents the taxon of the species.

**Figure 2 ijms-26-00020-f002:**
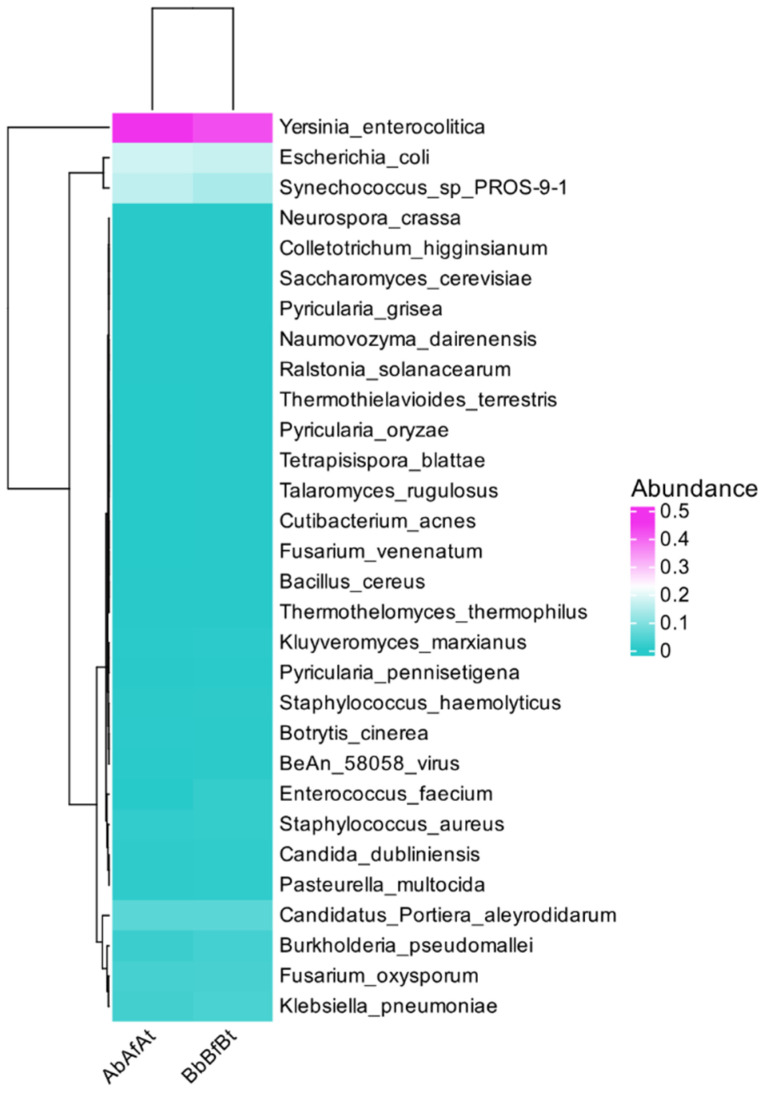
Species abundance heatmap. Each row represents a species, and each column represents a patient (A and B). The top 30 species of microorganisms with the highest abundance are shown in the diagram. The species clustering tree is on the left. The closer the branches of the clustering tree, the more similar they are. The square’s color represents the species’ abundance in the samples. The scale bar (to the heatmap’s right) shows a particular species’ abundance.

**Figure 3 ijms-26-00020-f003:**
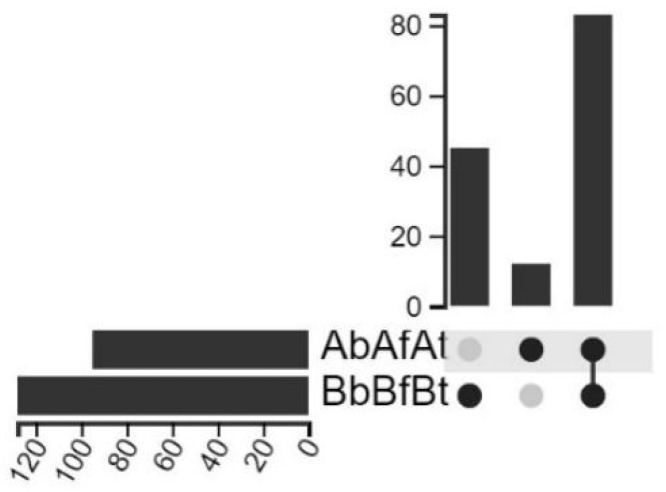
The UpSetR diagram shows similarities and differences in species composition identified in samples taken from both patients. The bar chart on the left represents the number of all species in specimens from patients A and B. The bar chart on the right represents the number of species in specimens unique to individual patients (B; the first bar, A; the second bar) and the number of species in specimens common to both patients. Samples were analyzed using relative abundance filtering/mean abundance.

**Figure 4 ijms-26-00020-f004:**
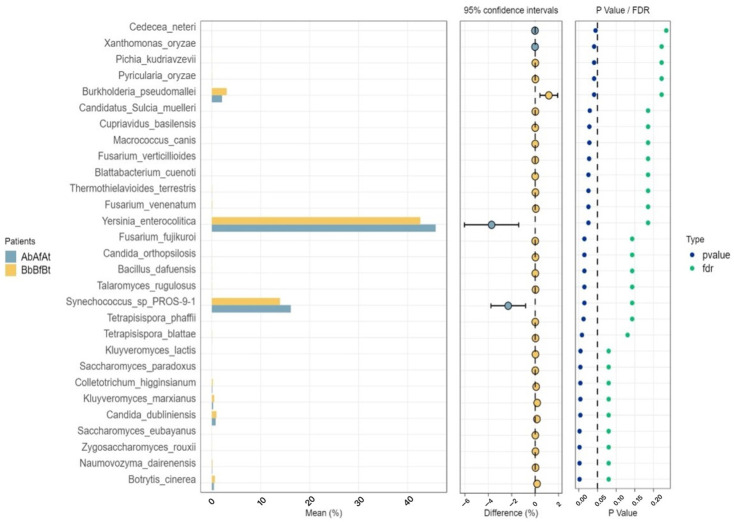
STAMP expanded histogram showing analysis of differences between patients in terms of species abundance. In the histogram on the left, the ordinate represents different species, the abscissa represents average group abundance (%), and the color of the columns indicates patients (A and B). The dots in the scatter plot on the right indicate test results with significant differences (the *p*-value), and FDR is the false discovery rate. The middle area shows the 95% confidence interval for the statistical test of the difference in abundance between the two patients. The position of the dots indicates the average value of the abundance differences, the color corresponds to that of the group with higher abundance, and the boundary of the line segment connected by dots represents the confidence limit.

**Figure 5 ijms-26-00020-f005:**
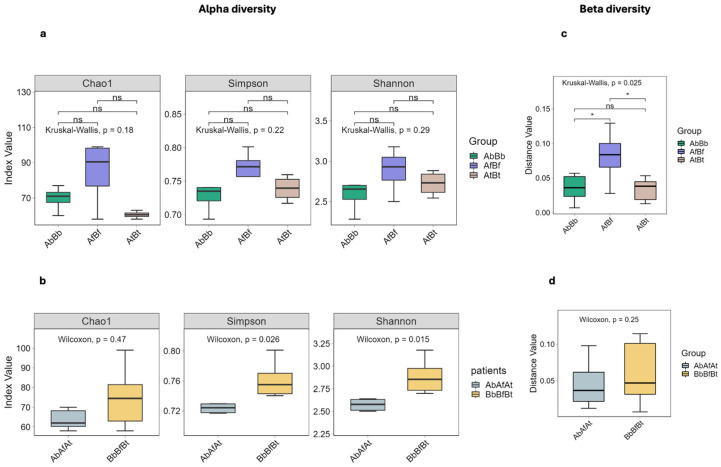
Species alpha and beta diversity boxplots. Each boxplot represents a diversity index. The abscissa and different boxes indicate groups of samples; the ordinates indicate the index values. The upper and lower edges of the box represent the first quartile and lower quartile. The horizontal line within the box represents the median of distances, and the ends of the straight line above and below the box represent the maximum and minimum distances, respectively; *p* < 0.05 indicates a significant difference between the groups of samples; * indicates *p* < 0.05; ns; indicates *p* > 0.05). (**a**) Species alpha diversity boxplot. The Kruskal–Wallis H test was used to determine the variability between groups of samples. No substantial differences in Chao1, Simpson and Shannon indices were observed. (**b**) Species alpha diversity boxplot. The Wilcoxon test was used to determine the variability between samples taken from individual patients (A and B); comparison between patients A and B showed substantial differences in Simpson and Shannon indices. This indicates a significant diversity of microorganisms and the richness and uniformness of samples taken from patient B. (**c**) Species beta diversity boxplot. The Kruskal–Wallis H test was used to determine the distance between samples. Beta diversity results indicate a difference in species composition between blood samples and synovial fluid samples and between fluid samples and synovial tissue samples. (**d**) Species beta diversity boxplot. The Wilcoxon test was used to determine variability between samples taken from individual patients (A and B). No significant difference in beta indices between patients was observed.

**Figure 6 ijms-26-00020-f006:**
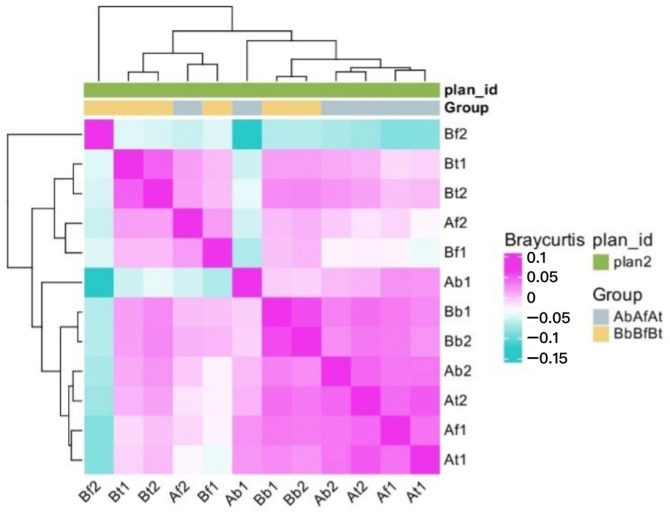
Sample similarity heatmap. Each row and column represents a sample (12 samples), and the color of the square represents the similarity index (Bray–Curtis anisotropy index) value or distance between two samples. Magenta reflects a smaller dissimilarity coefficient, and cyan represents a larger dissimilarity coefficient (the minimum similarity) in the heatmap graph. The left and top are sample cluster maps; the closer the branches, the more similar the samples.

**Figure 7 ijms-26-00020-f007:**
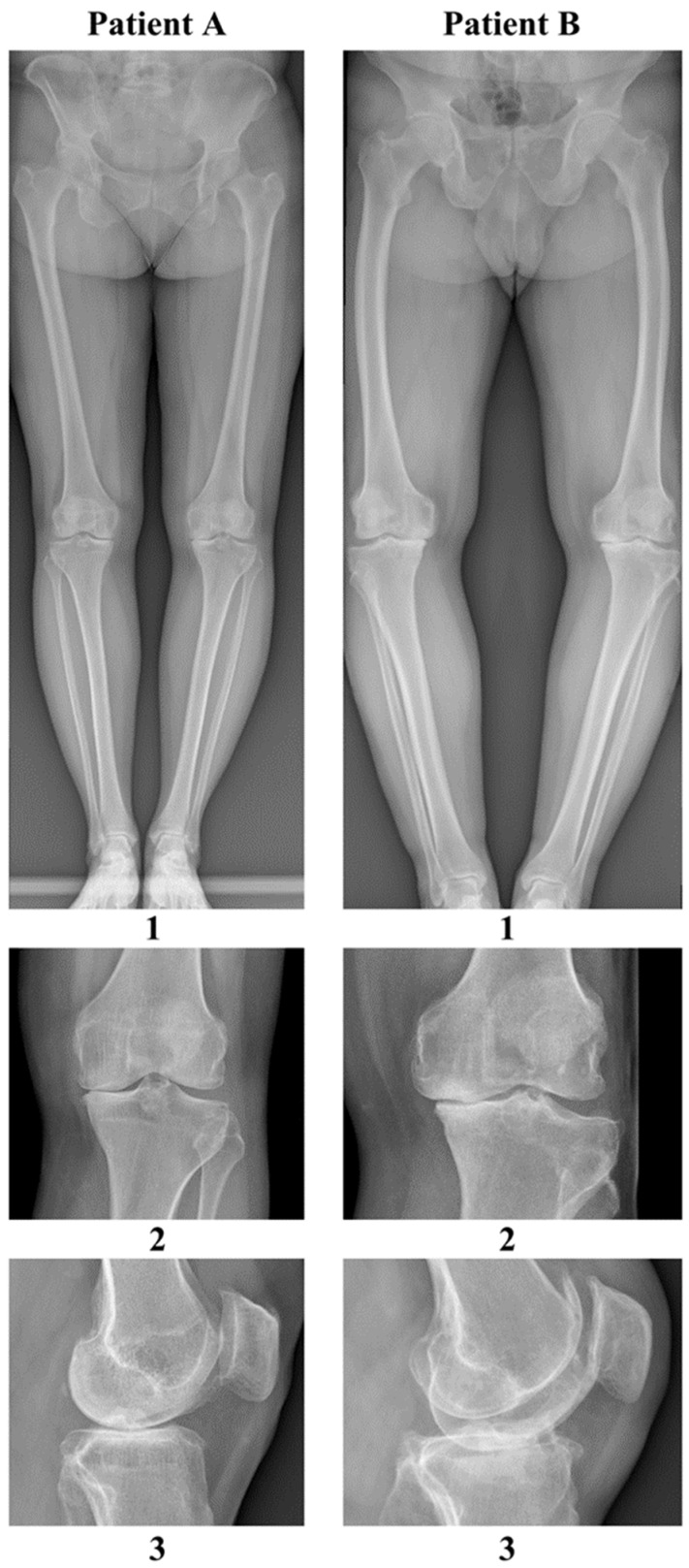
Pre-operative radiographs of the knees of patient A and patient B. Key: 1. Axial AP view of lower limbs, 2. AP view of left knee, 3. Lateral view of the left knee.

**Table 1 ijms-26-00020-t001:** Patient characteristics and medical history.

Feature	Patient A	Patient B
Gender	female	male
Height	165 cm	170 cm
Weight	77 kg	96 kg
BMI	28.0 kg/m^2^	33.6 kg/m^2^
Blood type	A Rh (−) (no antibodies)	B Rh (+) (no antibodies)
Age	69	61
ASA scale	2	2
Occupation	office worker	farmer (farm work, contact with domesticated ruminants)
Chronic diseases	hypertension overweight restless legs syndrome, multilevel spondyloarthritis positive tuberculin skin test during childhood	hypertension, obesity class 1, recurrent urinary tract infections, during childhood
Chronic medications	ropinirole rosuvastatin lacidipine, valsartan mianserin, paracetamol + tramadol	amlodipine + telmisartan acetylsalicylic acid potassium aspartate magnesium aspartate
Previous surgeries and procedures	tonsillectomy (age 15) appendectomy (age 16) ovarian cyst removal (age 46) dental prosthetic implantation (age 64) decompression of stenosis in the lumbar spine (age 68)	right-sided inguinal hernia (age 58)
Allergies	none	household dust trimethoprim–sulfamethoxazole

## Data Availability

The raw data supporting the conclusions of this article will be made available by the authors on request.

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
