# Peer review of "Exploring the Link Between Infections and Primary Osteoarthritis: A Next-Generation Metagenomic Sequencing Approach"

_ijms, 2024, doi:10.3390/ijms26010020_

Round 1
Reviewer 1 Report
Comments and Suggestions for Authors
The authors of this pilot study explore the potential link between knee osteoarthritis and colonization by resident flora using new metagenomic next-generation sequencing methodologies. They analyze blood, synovium, and synovial fluid from only two patients undergoing bilateral total knee replacement for idiopathic osteoarthritis. The authors report that the microbial flora most associated with osteoarthritic pathology are Yersinia enterocolitica, Escherichia coli, and Synechococcus. They conclude that mNGS has the potential to enhance understanding of the genetic, inflammatory, and infectious factors in the development of knee osteoarthritis.
Overall, the study is highly interesting, exploring an uncharted area of the etiopathogenesis of knee osteoarthritis. However, I question whether drawing conclusions from only two patients might be overly ambitious and potentially biased. How can the authors exclude false positives and false predictive values from their analyses? A method capable of identifying bacterial DNA present at less than 1% could overestimate the role of this factor, especially without corroborative tests like traditional cultures or polymerase chain reaction. The inability to obtain samples from healthy individuals adds the risk of not having true negatives to optimize the test's specificity, sensitivity, accuracy, and predictive value.
Therefore, I recommend conducting the same study with a larger sample size (considering the hospital performs 300 knee replacements annually, this should not be difficult) and, if possible, including a control group (patients treated arthroscopically for traumatic meniscal or ligament injuries could be used). An a priori estimation of the number of patients needed for enrollment would also be beneficial.
Author Response
Comment 1: However, I question whether drawing conclusions from only two patients might be overly ambitious and potentially biased.
Response 1: Thank you for pointing this out. We agree with this comment.
The analysis of samples from only two patients does not allow for definitive conclusions at this stage. The pilot study was designed to guide further research that would include additional methodologies and a larger group of carefully selected patients. The aim of this pilot study was to determine whether microbial DNA is present in the tissues of patients with primary osteoarthritis. The results obtained indicate the need for continued research, which will enable us and other researchers to draw more reliable conclusions.
We have added the following suggested changes on page 3, in the 'Objectives' section, lines: 106 – 108 and on page 3, in the ' Materials and Methods ' section, lines: 114 – 115.
We have emphasised that our survey is a pilot study in section 2. Materials and methods and in section 4.1 The limitation of the study.
Comments 2: How can the authors exclude false positives and false predictive values from their analyses? A method capable of identifying bacterial DNA present at less than 1% could overestimate the role of this factor, especially without corroborative tests like traditional cultures or polymerase chain reaction.
Response 2: Thank you for this question. We understand these doubts; they should always be taken into account when using highly sensitive techniques for the identification of microorganisms. When using mNGS, the interpretation of results must be approached very carefully. The very small amount of microbial DNA relative to the total DNA present in the tissue sample requires appropriate interpretation and the use of rigorous quality control methods at each stage of the assay.
We have added the following suggested changes on page 12, in the ' Discussion ' section, lines: 359 – 370 and on page 14 in the ' Discussion ' section, lines: 475 – 480.
In order to reduce doubts about, among other things, contamination, we have tried to describe the procedures used in detail in the Materials and Methods section and in the supplementary materials.
Comments 3: Therefore, I recommend conducting the same study with a larger sample size (considering the hospital performs 300 knee replacements annually, this should not be difficult) and, if possible, including a control group (patients treated arthroscopically for traumatic meniscal or ligament injuries could be used). An a priori estimation of the number of patients needed for enrollment would also be beneficial.
Response 3: Thank you for pointing this out. We agree with this comment.
With the results obtained, we can now plan the next targeted study accordingly. This was difficult at the stage of the preliminary study. Firstly, we plan to include additional methods to confirm the results of the metagenomic analysis (PCR and serology).
Secondly, using the data from this pilot study, we plan to conduct a subsequent study with at least 50 patients with primary osteoarthritis and a control group. The control group will include intra-operative taken tissue and blood samples from patients with traumatic knee injury plus blood samples from healthy volunteers. The inability to evaluate a larger patient and control group at this stage is due to the high cost of metagenomic NGS sequencing.
Thus, as we could not predict the results and, therefore, also taking into account the invasive nature of the procedures for collecting biological material for analysis, we were not able to confirm them adequately with other tests but tried to eliminate the confounders.
We believe that releasing the results of our pilot study can help share our findings with other researchers and attract funding for an extended research project in this area.
We have added the following suggested changes on page 15, in the ' Limitations of the Study ' section, lines: 500 – 503.
Reviewer 1 response summary
We fully acknowledge the preliminary nature of our study. Given that we analyzed samples from only two patients, we recognize that our conclusions must be viewed as exploratory rather than definitive. Our primary aim was to determine whether microbial DNA could be detected in the tissues of patients with primary osteoarthritis and to guide future, more comprehensive research efforts.
We agree that increasing the sample size, particularly since our hospital performs around 300 knee replacements annually, and including a control group, such as patients with traumatic knee injuries, would greatly enhance the reliability and relevance of our findings. We also acknowledge the importance of confirming our results with additional methods, including PCR, serology, and possibly more traditional testing approaches, to ensure we can accurately interpret the significance of the detected microbial DNA.
In conducting our study, we employed strict quality control measures at every stage—sampling, DNA isolation, library construction, and sequencing—under controlled environmental conditions. We used the DNBSEQ platform known for its low error and misassignment rates, and we set a threshold (10% of non-human reads) to minimize the likelihood of false positives. While we recognize that interpreting data from highly sensitive metagenomic sequencing methods requires caution, we believe that these procedural safeguards help mitigate potential contamination and overestimation of microbial involvement.
Moving forward, we plan to leverage our pilot results to secure the support and funding necessary for a more extensive study. We envision a larger cohort (at least 50 patients), the inclusion of a well-defined control group, and the use of confirmatory diagnostic methods to build a stronger evidence base. We believe sharing our initial findings is a crucial first step toward improving understanding, generating research interest, and ultimately advancing our knowledge of the complex role that microbial flora may play in the pathogenesis of knee osteoarthritis.
Reviewer 2 Report
Comments and Suggestions for Authors
General characteristics and evaluation of the reviewed article:
The article Exploring the link between infections and primary osteoarthritis: a next-generation metagenomic sequencing approach presents an innovative pilot study investigating the relationship between microorganisms and primary knee osteoarthritis (OA) using metagenomic next-generation sequencing (mNGS). The authors analyzed synovial membrane, synovial fluid, and blood samples from two patients undergoing knee arthroplasty. Their findings revealed a predominance of microorganisms from the Enterobacterales order, with Yersinia enterocolitica being the most abundant species, followed by Escherichia coli and Synechococcus sp., the latter indicating potential environmental exposure to microbial toxins.
The study highlights a novel perspective on OA, suggesting it involves not only degenerative processes but also genetic, inflammatory, and infectious components. This approach has the potential to reshape our understanding of OA pathogenesis.
However, the study is limited by its small sample size, lack of controls, and absence of functional analyses to clarify the role of identified microbes. Future research should address these limitations by including larger cohorts, control groups, and mechanistic studies to explore the functional impact of microorganisms on OA progression.
In conclusion, this research underscores the potential of mNGS to uncover microbial contributions to OA and calls for a multidisciplinary approach to better understand the disease’s complexity. While preliminary, the findings offer a promising direction for future investigations into novel diagnostic and therapeutic strategies.
The article is interesting, addresses a timely and important topic and definitely fits the scope of the journal. It is written generally correctly and requires only minor corrections and additions before further processing and acceptance for publication. Below are my points and detailed comments.
Minor comments:
Please provide a more comprehensive discussion of osteoarthritis in the first paragraph. This will create a stronger introduction to the topic and emphasize the importance of the issue. The incidence of osteoarthritis is affected by various factors such as occupation, sports participation, musculoskeletal injuries, obesity, and gender. Information about these factors, along with relevant literature, should be included in the first paragraph of the introduction. I recommend adding the following references to this section:
https://doi.org/10.3390/healthcare12161648
DOI: 10.1056/NEJMcp1903768
Figure 3 and 4 need to be improved, fonts need to be adjusted to meet editing requirements and resolution needs to be improved. Please improve before resubmitting.
The inclusion of only two patients severely limits the generalizability and statistical power of the findings. This sample size is insufficient to establish any causal or correlative relationships. Please detail the descriptions.
The absence of healthy controls or patients with other joint conditions reduces the ability to discern whether the identified microorganisms are specific to primary knee OA. Please detail the descriptions.
Despite precautions, the study acknowledges that mNGS is highly sensitive to contamination. Many of the identified microorganisms could originate from environmental, procedural, or laboratory contamination, making their relevance to OA unclear. Please complete the descriptions with references.
he study does not perform functional analyses to determine whether the identified microorganisms actively contribute to OA pathogenesis or if they are merely bystanders.
The authors suggest a potential link between Y. enterocolitica and OA but fail to provide sufficient evidence or a clear mechanism. Historical reports of reactive arthritis caused by Y. enterocolitica do not directly support a link to primary OA. Please correct and expand the descriptions in the discussion.
The presence of microbial DNA in tissues does not necessarily imply infection or pathogenicity. The article lacks evidence that these microorganisms are metabolically active or contribute to disease progression.
The differences in microbial diversity between the two patients are noted, but their significance is not explored in depth, leaving questions about inter-individual variability unanswered.
In the final part of the discussion, please describe in more detail the limitations of the proposed method, the simplifications used, and a proposal for solving them in the authors' further planned future research.
I congratulate the authors on the interesting paper and wish them further success.
Author Response
Comments 1: Please provide a more comprehensive discussion of osteoarthritis in the first paragraph. This will create a stronger introduction to the topic and emphasize the importance of the issue. The incidence of osteoarthritis is affected by various factors such as occupation, sports participation, musculoskeletal injuries, obesity, and gender. Information about these factors, along with relevant literature, should be included in the first paragraph of the introduction. I recommend adding the following references to this section: https://doi.org/10.3390/healthcare12161648 and DOI: 10.1056/NEJMcp1903768
Response 1: Thank you for pointing this out. We agree with this comment.
We have added suggested information to the ‘Introduction’ section, page number 2, lines 36 – 43, 49 – 53, 61 – 64 and references number 3, and 4.
Comments 2: Figure 3 and 4 need to be improved, fonts need to be adjusted to meet editing requirements and resolution needs to be improved. Please improve before resubmitting.
Response 2: Thank you for pointing this out. We agree with this comment.
We have made the suggested changes to Figures 3 and 4. ‘Microbial Communities’ section, pages: 9 and 10.
Comments 3:The inclusion of only two patients severely limits the generalizability and statistical power of the findings. This sample size is insufficient to establish any causal or correlative relationships. Please detail the descriptions.
Response 3: Thank you for pointing this out. We agree with this comment.
The analysis of samples from only two patients does not allow for definitive conclusions at this stage. The pilot study was designed to guide further research which would utilise additional methodologies and a larger group of carefully selected patients as well as a control group. The aim of this pilot study was to determine whether microbial DNA is present in the tissues of patients with primary osteoarthritis. The results obtained indicate the need for continued research, which will enable us and other researchers to draw more reliable conclusions.
We have included additional details characterising the pilot study in the ‘Objective’, ‘Materials and Methods’, the ‘Limitations of the Study’ sections.
1.1. Objective section, page 3, lines: 106 - 108
- Materials and Methods section, page 3, lines 114 - 115
4.1. The Limitations of the Study section, page number 15, lines: 500 – 503.
Comments 4: The absence of healthy controls or patients with other joint conditions reduces the ability to discern whether the identified microorganisms are specific to primary knee OA. Please detail the descriptions.
Response 4: We agree with this comment. Using the data from this pilot study, we plan to conduct a subsequent study with at least 50 patients with primary osteoarthritis and a control group. The control group will include intra-operative taken tissue and blood samples from patients with traumatic knee injury plus blood samples from healthy volunteers. The inability to evaluate a larger patient and control group at this stage is due to the high cost of metagenomic NGS sequencing.
We have added the following suggested changes on page 15, in the ' Limitations of the Study ' section, lines: 500 – 503.
Comments 5: Despite precautions, the study acknowledges that mNGS is highly sensitive to contamination. Many of the identified microorganisms could originate from environmental, procedural, or laboratory contamination, making their relevance to OA unclear. Please complete the descriptions with references.
Response 5: Thank you for pointing this out. We agree with this comment. We are aware of all the limitations.
We have completed the ‘Discussion’ section; page nr 12, lines 359 – 370, page 14, lines 475 – 480, and added references nr 18 and 47.
Comments 6: The study does not perform functional analyses to determine whether the identified microorganisms actively contribute to OA pathogenesis or if they are merely bystanders.
Response 6: This was for a very mundane reason. The cost of doing a metagenomic analysis is still very high. In a situation where we did not know what identification results to expect, we deliberately decided against functional analysis. We now conclude that the results of this functional analysis would also be difficult to interpret, raising questions related to an inflammatory rather than infectious process in OA.
We have added the following suggested changes on page 15, in the ' Limitations of the Study ' section, line: 513.
Comments 7: The authors suggest a potential link between Y. enterocolitica and OA but fail to provide sufficient evidence or a clear mechanism. Historical reports of reactive arthritis caused by Y. enterocolitica do not directly support a link to primary OA. Please correct and expand the descriptions in the discussion.
Response 7: Thank you for pointing this out. We agree with this comment.
We have deleted one sentence in the ‘Discussion’ section on page 13, lines 392-394, and we have included this suggestion in the ‘Discussion’ section on pages 13 – 14, lines: 428 – 332.
Comments 8: The presence of microbial DNA in tissues does not necessarily imply infection or pathogenicity. The article lacks evidence that these microorganisms are metabolically active or contribute to disease progression.
Response 8: Thank you for pointing this out. We agree with this comment. Of course, the detection of genetic material at a disease site alone does not prove an aetiological link between the microorganism and the pathology but is a trigger for further research. This is the basic principle for interpreting any microbiological examination.
We have added the following suggested changes on pages 13 – 14, in the ' Discussion ' section, lines: 428 – 332.
Comments 9: The differences in microbial diversity between the two patients are noted, but their significance is not explored in depth, leaving questions about inter-individual variability unanswered.
Response 9: Thank you for pointing this out. We agree with this comment.
We have been rather cautious in interpreting the differences in the results obtained in the study of samples from two patients. We only pointed out differences in relative abundance. We felt that any attempt at interpretation could only be speculative because of the small number of samples. The variation in the composition of the microbiome can depend on many factors, including past infections, other diseases, work, diet, lifestyle and so on. What these patients certainly have in common is the high percentage of Y. enterocolitica DNA in the samples examined.
Comments 10: In the final part of the discussion, please describe in more detail the limitations of the proposed method, the simplifications used, and a proposal for solving them in the authors' further planned future research.
Response 10: Thank you for pointing this out. We agree with this comment.
We have expanded the section ‘Limitations of the study’ in lines: 500 - 503, 505 - 507, 513 - 517 according to your suggestions.
First, considering this is a pilot study designed to guide further research, careful selection of the small sample number included represents a limitation of this study. Besides this aspect, there is an acknowledged addition in the future of samples from the healthy-without clinical or radiological signs of OA-and having taken an yersiniosis history, as well as the completion of at least 50 cases with OA.
These findings are not confirmed by direct confirmation of bacteria in clinical samples. Finally, mNGS is an advanced technique that correctly identifies the DNA of etiological agents responsible for the disease. However, with the high sensitivity of this method, the results are uninformative and poorly interpretable at this level of knowledge.
Further studies will add the inclusion of real-time PCR technique parallel for the detection of specific microbial sequences in clinical samples, determination of threshold cycle. These will be done along with mNGS analysis including functional analysis. Animal models can be another useful tool in the investigation of OA pathogenesis. This way of infective induction and inflammation will help to define not only intracellular changes and processes but also further establish milestones in the understanding of such a complex pathology.
Round 2
Reviewer 1 Report
Comments and Suggestions for Authors
The authors have addressed my comments, clarifying the study's goal as a preliminary exploration of using mNGS to investigate the etiopathogenetic mechanisms of knee osteoarthritis, particularly the role of resident microbial flora in the knee (traditionally considered sterile). While the experimental design, methodology, sample size, and results remain unchanged, the authors plan to continue with larger cohorts based on these initial findings. Although the study's quality hasn't improved, framing it as a methodological exploration of this new technique in an under-researched area makes the objectives more focused. I recommend revising the conclusions in both the abstract and manuscript to: "This article provides an initial exploration of mNGS use to study the etiopathogenetic mechanisms of knee OA. While our analysis identified bacterial DNA, particularly from Yersinia, further cross-sectional studies in larger populations with and without OA are needed to determine the role of these agents in OA pathogenesis."
Author Response
Comments 1: The authors have addressed my comments, clarifying the study's goal as a preliminary exploration of using mNGS to investigate the etiopathogenetic mechanisms of knee osteoarthritis, particularly the role of resident microbial flora in the knee (traditionally considered sterile). While the experimental design, methodology, sample size, and results remain unchanged, the authors plan to continue with larger cohorts based on these initial findings. Although the study's quality hasn't improved, framing it as a methodological exploration of this new technique in an under-researched area makes the objectives more focused. I recommend revising the conclusions in both the abstract and manuscript to: "This article provides an initial exploration of mNGS use to study the etiopathogenetic mechanisms of knee OA. While our analysis identified bacterial DNA, particularly from Yersinia, further cross-sectional studies in larger populations with and without OA are needed to determine the role of these agents in OA pathogenesis.
Response 1:
We fully agree with the reviewer's comment and accept the proposed changes to the text. We have therefore changed the text in accordance with the suggestions.